# Laboratory validation of a simplified DNA extraction protocol followed by a portable qPCR detection of *M. tuberculosis* DNA suitable for point of care settings

**Tainá dos Santos Soares**[1,2], **Graziele Lima Bello**[1], **Ianca Moraes dos Santos Petry**[1], **Maria Rita Castilho Nicola**[1], **Larissa Vitoria da Silva**[2], **Regina Bones Barcellos**[2], **Joana Morez Silvestri**[3], **Maria Lucia Rossetti**[1], **Alexandre Dias Tavares Costa**[4]*

1 Laboratório de Biologia Molecular, ULBRA, Canoas, RS, Brasil, 2 Centro de Desenvolvimento Científico e Tecnológico (CDCT)–Centro Estadual de Vigilância em Saúde (CEVS)–Secretaria do Estado do Rio Grande do Sul (SES/ RS), Porto Alegre, RS, Brasil, 3 Hospital Universitário de Canoas (HU), Canoas, RS, Brasil, 4 Instituto Carlos Chagas (ICC), Fiocruz, Curitiba, PR, Brasil

* alexandre.costa@fiocruz.br

**Data Availability Statement:** All relevant data are within the paper and its Supporting Information files.

## Abstract

Tuberculosis, caused by *Mycobacterium tuberculosis*, is a treatable and curable disease, and yet remains one of the leading causes of death worldwide. Diagnosis is essential to reducing the number of cases and starting treatment, but costly tests and equipments that require complex infrastructure hamper their widespread use as a tool to contain the disease in vulnerable populations as well countries lacking resources. Therefore, it becomes necessary to develop new technological approaches to molecular methods as well as screening tests that can be rapidly conducted among people presenting to a health facility to differentiate those who should have further diagnostic evaluation for TB from those who should undergo further investigation for non-TB diagnoses. The present study aimed to evaluate two experimental DNA extraction methods from clinical samples (FTA card versus sonication) followed by analysis in a portable qPCR instrument (the Q3-plus). The FTA card-based protocol showed 100% sensitivity and specificity, while the sonication protocol showed 80% sensitivity and 89% specificity when compared to the traditional gold standard culture. The portable protocol, comprised by the FTA card method and the portable instrument Q3-Plus, showed sensitivity and specificity of 92% and 61%, respectively, when compared to culture, and 75% and 81%, respectively, when compared to the standard TB case classification. The ROC curve showed an AUC of 0.78 (p<0.001) for the portable protocol and 0.93 (p<0.001) for the GeneXpert Ultra. The limit of detection (LOD) for *Mycobacterium tuberculosis* (H37Rv strain) detection in spiked samples obtained using the portable protocol (FTA card and Q3-Plus) was 19.3 CFU/mL. As an added benefit, using the FTA card facilitates sample handling, transport, and storage. It is concluded that the use of the FTA card protocol and the Q3-Plus yields similar sensitivity and specificity as the gold standard diagnostic tests and case classification. We suggest that the platform is suitable to use as a point of care tool, assisting in the screening of tuberculosis in hard-to-reach or resource-limited areas.

**Funding:** This work was funded by grants from Conselho Nacional de Desenvolvimento Científico e Tecnológico (CNPq, https://www.gov.br/cnpq/pt-br) and Presidência da Fundação Oswaldo Cruz/ Vice Presidência de Pesquisa e Coleções Biológicas VPPCB/Fiocruz; Chamada – CNPq/Fiocruz Nº 30/ 2020– INOVA PROEP/PEC (CNPq 445954/2020-5) to ADTC and INCT-TB FAPERGS 17/1265 from CNPq and Fundação de Amparo à Pesquisa do Estado do Rio Grande do Sul (FAPERGS, https:// fapergs.rs.gov.br/inicial) to MLRR. TSS is a fellowship holder at CAPES (Fundação Coordenação de Aperfeiçoamento de Pessoal de Nível Superior, https://www.gov.br/capes/pt-br) (CAPES 001). ADTC is a CNPq Productivity Fellow (level 2). The funders played no role in the study design, data collection and analysis, decision to publish, or preparation of the manuscript. The funders played no role in the study design, data collection and analysis, decision to publish, or preparation of the manuscript.

**Competing interests:** The authors have declared that no competing interests exist.

# Introduction

Tuberculosis (TB) is one of the deadliest diseases in the world. In 2022, 10.6 million people developed active TB and 1.3 million people died [1]. In Brazil, in 2022, 5.845 people died and 84,858 were diagnosed positive in 2023 [2]. Traditional TB diagnosis occurs through sputum smear microscopy, an inexpensive and practical method, but with limited specificity and sensitivity. Despite their advantages, traditional diagnostic methods also have limitations. Mycobacterial growth culture is considered the gold standard method; however, although it achieves high sensitivity, it takes a long time to produce results (usually around 3 to 4 weeks). On the other hand, sputum smear microscopy has the advantage of good specificity but requires the presence of at least 5,000 to 10,000 bacilli per milliliter of sample for a positive result, thus showing lower sensitivity [3, 4].

Molecular tests emergence overcame some of the limitations presented by traditional tests, providing highly specific and sensitive diagnosis in just a few hours. Real-time PCR (qPCR) allows the amplification and quantification of nucleic acids and is currently the most used molecular technique for diseases diagnosis [5, 6]. In the Brazilian Unified Health System (SUS), the GeneXpert MTB/RIF® assay (Cepheid, USA) is the test of choice, providing results in up to 2 hours, and a bonus of being recommended by WHO since 2010. This test amplifies specific targets of *Mycobacterium tuberculosis (Mtb)* genome and the rifampin resistance determining region (RRDR) in *rpo*B gene [3, 5, 6]. However, GeneXpert MTB/RIF assay also has limitations, such as false-negative results, which were partially solved in the tests' newer versions, such as GeneXpert MTB Ultra®, which is more sensitive and less specific than the original version. The sensitivity and specificity of the two versions of GeneXpert vary depending on the clinical context and the population tested. For GeneXpert MTB/RIF in patients with positive bacilloscopy (pulmonary TB with a high bacterial load), the sensitivity is around 98%, with a specificity of 99%. For GeneXpert Ultra in patients with positive bacilloscopy, the sensitivity is like GeneXpert MTB/RIF, at around 98%, with a specificity of 96–98%." [6–8]. The implementation of the GeneXpert assay increased the number of positive diagnoses, but did not improve global case detection rates, as the equipment has high infrastructure requirements and costs, making it not viable for resource-limited areas where the TB burden is higher [3]. An alternative to increase the number of diagnoses is active search, using effective and accessible diagnostic tools with a rapid response time that can perform in resource-limited settings [4].

The implementation of tests where patients are treated ("point of care", or POC) allows for the referral of only confirmed cases to health centers [5, 6]. However, alternative screening methods are still needed to facilitate patients access to these technologies, reducing the initial barrier to obtain some medical evaluation and increasing the motivation of individuals with a higher probability of developing TB [7].

One of the challenges for molecular and portable screening tests is defining a process to obtain DNA that is simple and fast and does not require dangerous chemicals, complex biosafety infrastructure, manipulation, or storage [5, 7]. The method of choice must be fast, simple, low cost, and still provide a good quantity and quality (i.e. purity) of material. The extraction technique to be used must vary according to the sample type and the downstream applications, with varying steps of cell disruption, removal of lipids, proteins, and other nucleic acids, as well as purification and concentration of the targeted nucleic acids. For example, rupture of the cell wall can be performed mechanically, enzymatically, chemically, or by combinations of these techniques [9–11]. There are many methods of DNA extraction available as commercial kits for application in molecular analysis. Some laboratories, on the other hand, prefer to use in-house methods to reduce costs [11, 12]. Among the in-house methods already standardized, sonication deserves special consideration. This method is considered simple and

very effective for breaking cell walls, without the need of chemical reagents, enzymes, proteins, or substances that can compromise the integrity and detection of the sample's DNA [9, 10]. Alternatively, simple protocols using the FTA Elute Micro card (Whatman, USA) have been published [13–15]. Detergents are embedded in these FTA cards, making them multifunctional: in addition to being a transport and storage medium, detergents also help to solubilize cell membranes, thus releasing DNA in the extraction processes, which then binds to the cellulose matrix and is easily eluted with aqueous solutions whenever necessary. The sample attached to this card is less susceptible to contamination, keeping the material intact for years without the need for costly conservation strategies [9, 11, 12].

WHO recognizes the need for more sensitive and specific tests to improve the early diagnosis of TB, such as the development of portable POC devices, which stand out for their simplicity, accessibility, and portability [1, 5, 7]. Therefore, our study aimed to evaluate two DNA extraction protocols for the detection of *Mtb* from sputum, one sonication-based and the second FTA card-based [13]. The chosen protocol was further validated using a portable qPCR system, the Q3-Plus [16], in parallel to the routine diagnostic testing using the GeneXpert MTB/RIF/Ultra assay and culture.

## Materials and methods

### Samples origin

We used 127 sputum samples from patients over 18 years old, seeking treatment at the Department of Tisiology and Leprosy of the Health Department at the University Hospital of the Brazilian Lutheran Universtiy at Canoas (ULBRA), in the municipality of Canoas (south of Brazil), from October 1st 2018 to September 30th 2022. Data from the participants were accessed within the same dates (October 1st 2018 to September 30th 2022). This research was approved by the Ethics Committee of ULBRA (CAAE 0697116.7.0000.5349 to GLB) and by the Ethics Committee of FIOCRUZ (CAAE 4423120.0.1001.5248 to ADTC).

Written informed consent for sample collection as well as for data publication were obtained from all individual participants included in the study. No minors were included.

We used 29 samples, divided into two aliquots, totaling 58 samples to test protocol 1 and protocol 2, while 98 samples were used for validation of the simplified DNA extraction protocol followed by the portable instrument Q3-Plus. Samples with a minimum volume of 500 μL were characterized for presence or absence of *Mtb* by GeneXpert MTB/RIF assay. Protocols 1 and 2 were evaluated by GeneXpert MTB/RIF whereas GeneXpert Ultra and culture (Bactec MGIT) were used to validate the simplified protocol. After routine characterization, samples were sent to the Molecular Biology Laboratory at the Lutheran University of Brazil (ULBRA), where they were processed and submitted to the molecular tests objective of the present study.

### Standard clinical classification of tuberculosis cases ("case TB")

WHO defines a tuberculosis positive case as definitive when *Mtb* is identified in a patient's sample by either culture or a molecular assay [17, 18]. In countries that do not have the laboratory capacity to routinely identify *Mtb*, a pulmonary case with one or more initial sputum smear testing positive for acid-fast bacilli (AFB) is also considered a "definite" case, if there is a functional external quality assurance (EQA) system with blind rechecking. In this study, we classified all samples with positive culture or positive GeneXpert assay as positive TB cases according to SINAN (Brazilian Notifiable Diseases Information System) and the TB Site (Tuberculosis Special Treatment Information System) [17, 18]. Therefore, the diagnostic technique for these samples will be referred as "clinical case classification (or simply "case TB")".

## DNA extraction protocols

**Sonication-based.** Sputum samples were decontaminated with 4% sodium hydroxide (NaOH) in PBS (phosphate-buffered saline). A total of 500μL of sputum was transferred to a 1.5mL microtube and 4% NaOH was added, followed by vortex homogenization, and incubation at 37°C for 15min. Subsequently, it was centrifuged 3000g for 15min and the supernatant was discarded. The pellet was resuspended in 500μL of PBS and homogenized on a vortex. Next, the tube was incubated at 95°C for 20min. After this step, it was placed in the sonication bath for 15min, and then centrifuged for 5min at 3000g. Finally, 100μL of the supernatant (containing the DNA) was transferred to a new microtube and frozen at −20°C until use [19].

**FTA card-based.** DNA extraction with FTA® Elute Micro card (Whatman, USA) was performed as described by Ali *et al* 2020 [14], with minor modifications. Briefly, an aliquot of the sample was mixed with a solution containing 6 M guanidine isothiocyanate and 0.5 M EDTA (ratio of 400 μL denaturing solution per 1000 μL sample) and 20 μl of Proteinase K (25 mg/ml, Roche Diagnostics, Germany). Next, the sample was vigorously shaken in vortex for 20–30 seconds. The mixture was evenly applied to a FTA® Elute Micro card using a plastic Pasteur pipette, and airdried for at least 1 hour at room temperature. From this point on, the card can be used directly for further processing or for sample storage. For DNA extraction, a 6 mm diameter disc was punched out and placed in a 1.5 mL microtube. TE buffer pH 8.0 (500 μL) was added to the disc-containing tube which was then vortexed and incubated at 95ºC for 5 min. The tube was then centrifuged for 1 minute at maximum speed, and 50–100 μL of the supernatant was transferred to a new tube and stored at -20ºC until qPCR amplification [14].

## Real-time PCR

DNA amplification and detection of *Mycobacterium tuberculosis* were performed using real-time PCR (qPCR) in a Step One Real-Time PCR system (AB Applied Biosystems, USA), with oligonucleotides targeting the *IS6110* genomic marker, which is present in multiple copies within the *M. tuberculosis* genome and is exclusively found in the *Mycobacterium* complex [20, 21]. (Forward 5′-CAGGACCACGATCGCTGAT-3′ and reverse 5′-TGCTGGTGGTCCGAA GC-3′) and the probe (5′- FAM-TCCCGCCGATCTCG- HQI-3′) were used. Each reaction had a final volume of 10 μL, containing 0.5 μL oligomix (20X primer and probe solution), 3.5 μL PCR mix (Kapa Probe Fast qPCR Mastermix), 3.5 μL ultrapure water and 2.5 μL extracted DNA. The reference strain H37RV (10 ng/μL) was used as a positive control (PC) and ultrapure water was used as a negative control (NC). Amplification conditions were as follows: 50ºC for 2 minutes, followed by 10 minutes at 95ºC, and 95ºC for 15 and annealing and amplification at 60ºC for 1-minute seconds for a total of 40 cycles [22].

DNA amplification and detection of the *IS6110 M. tuberculosis* genomic marker using the portable Q3-Plus instrument (ST Microelectronics, Italy) was performed using the same reagents described above, but for a final reaction volume of 5 μL. The reaction contained 0.25 μl of *IS6110* oligomix, 2.5 μl of qPCR master mix, 0.75 μl of ultrapure water, and 1 μl of extracted DNA. The amplification conditions were as follows: 95ºC for 10 minutes, followed by 45 cycles of 95ºC for 15 seconds and 60ºC for 60 seconds. The optical parameters for the FAM channel in the Q3-Plus system were exposure time of 1 second, LED power of 3, and analog gain of 15, while for the HEX channel the optical parameters were exposure time of 2 seconds, LED power of 10, and analog gain of 15. The reaction was supplemented with 0.5 μl oligonucleotides for detection of the human 18S rRNA gene (Forward 5′ TGCGAATGGCTCA TTAAATC 3′, Reverse 5′ CGTCGGCATGTATTAGCTCT, and HEX-probe TGGTTCCTTTG

GTCGCTCGCT-BHQ1), which was used as an internal control [14]. In both instruments, baseline and threshold were set to automatic.

## Limit of detection (LOD$_{95\%}$): Colony forming units in parallel to qPCR

Reference strain *Mtb* H37Rv colonies cultivated in the Ogawa-Kudoh medium were collected and homogenized with glass beads. The turbidity of the bacterial solution was compared to the turbidity of the McFarland number 1 standard. Subsequently, ten-fold serial dilutions were performed from the concentrated cell suspension, and tubes containing 500 μL of mucin suspensions (20% v/w) were spiked with 30 μL of each dilution. The whole volume was equally distributed onto individual FTA Elute Micro cards and the simplified DNA extraction protocol was performed ("FTA card-based protocol" above). This procedure was performed in duplicate for each cell suspension concentration. The same procedure was performed in parallel and spiked samples were plated in petri dishes containing 7H10/OADC medium, in duplicates. Colonies were counted using a semi-quantitative scale, according to the current Brazilian standards [18].

## Statistical analysis

Reactions on the Step One instrument were performed in triplicates, and reactions on the Q3-Plus instrument were performed in duplicates (optimization protocols) or quadruplicates (patient samples). The Cohen's kappa coefficient was calculated between the Q3-Plus or Step One results obtained with the FTA card-based protocol and the results obtained with GeneXpert or culture, using these latter methods as the gold standard with a 95% CI (confidence interval). The Kappa (K) agreement force was interpreted as follows: Strength of Agreement < 0.00 (Poor), 0.00–0.20 (Slight), 0.21–0.40 (Fair), 0.41–0.60 (Moderate), 0.61–0.80 (Substantial) and 0.81–1.00 Almost Perfect) [12]. Sensitivity and specificity were evaluated about the tuberculosis case definition by the attending physician. Student´s t-test with a significance level of 0.05 was used to evaluate the difference between results obtained with the Step One and the Q3-Plus system. The LOD$_{95\%}$ for the DNA extraction protocols was estimated by qPCR in parallel to colony growth using a Probit regression.

All data were analyzed using Statistical Package for the Social Sciences (SPSS), version 21.0. All relevant data are within the manuscript and its Supporting Information files.

## Results and discussion

### Sample analysis

Sputum samples from 29 patients were characterized by culture and GeneXpert MTB/RIF test before analysis by the two experimental DNA extraction protocols. Data from GeneXpert MTB/RIF test revealed that 14/29 (48.2%) were positive and 15/29 (51.7%) were negative samples, while culture analysis showed that 11/29 (38%) were positive and 18/29 (62%) were negative samples.

Samples were also aliquoted for analysis by both experimental extraction methods (Sonication- and FTA card-based). The presence of *Mtb* was confirmed in the final eluate of each protocol by qPCR (Fig 1). The sonication protocol showed that 10/29 (34,4%) were positive and 19/29 (65.5%) were negative samples, while the FTA card protocol revealed that 11/29 (38%) were positive and 18/29 (62%) were negative samples. Overall, 24/29 samples agreed as positive or negative between all four techniques. These results are also shown as a direct comparison of the results obtained by each experimental protocol per sample (Fig 2 and Table 1). The data show that Cts obtained by the sonication protocol are lower than those obtained by the FTA

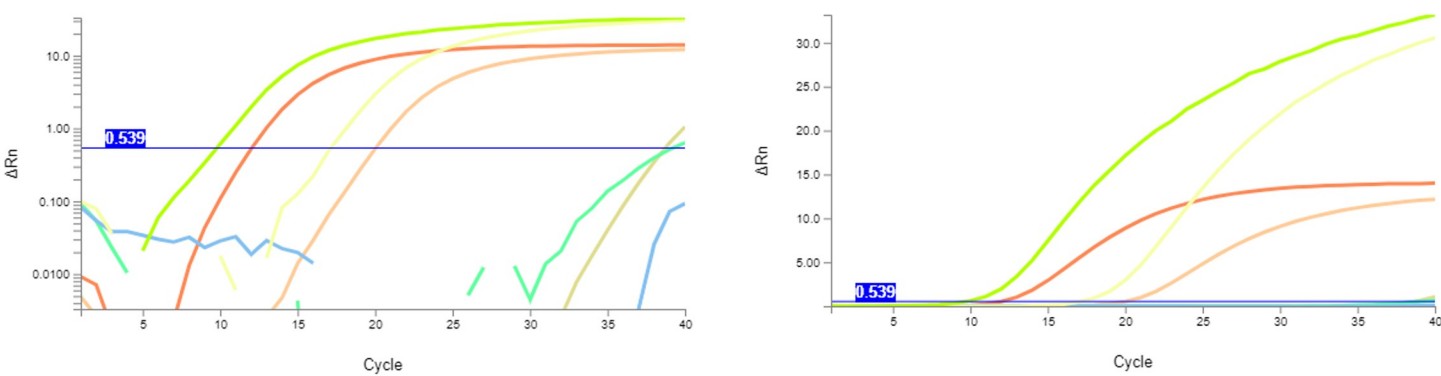

**Fig 1.** Representative curves of qPCR detection of the *Mtb* by the StepOne (panel A) or the Q3-Plus (panel B) instruments. Representative curves for qPCR detection of the *Mtb* genomic marker *IS6110* in samples processed by the sonication or the FTA card experimental protocols. "PC" and "NC" represent the positive control (25 ng/μL of DNA extracted from H37Rv MTB cells) and the negative control (TE pH 8.0). Traces "a" and "b" were obtained with sonication and FTA card protocols using sample #4, traces "c" and "d" were obtained with sonication and FTA card protocols using sample #13, and traces "e" and "f" obtained with sonication and FTA card protocols using sample #14, respectively. Traces are representative of at least three independent experiments for each protocol.

card protocol by an average of– 2.5 Cts, but with higher variation (26.7 ± 7.50 versus 29.2 ± 4.52). However, the FTA card protocol yielded more positive samples than sonication.

When the results obtained by both experimental protocols were compared with the GeneXpert MTB/RIF results, the FTA card protocol detected 11 *Mtb* samples with GeneXpert assay detecting an additional 3 samples (#16, #21, and #22). The sonication protocol detected 9 samples with the GeneXpert assay detecting 4 more samples (#15, #20, #21, and #28). Compared to the culture, two samples (#20 and #28) presented a false negative result and one (#22)

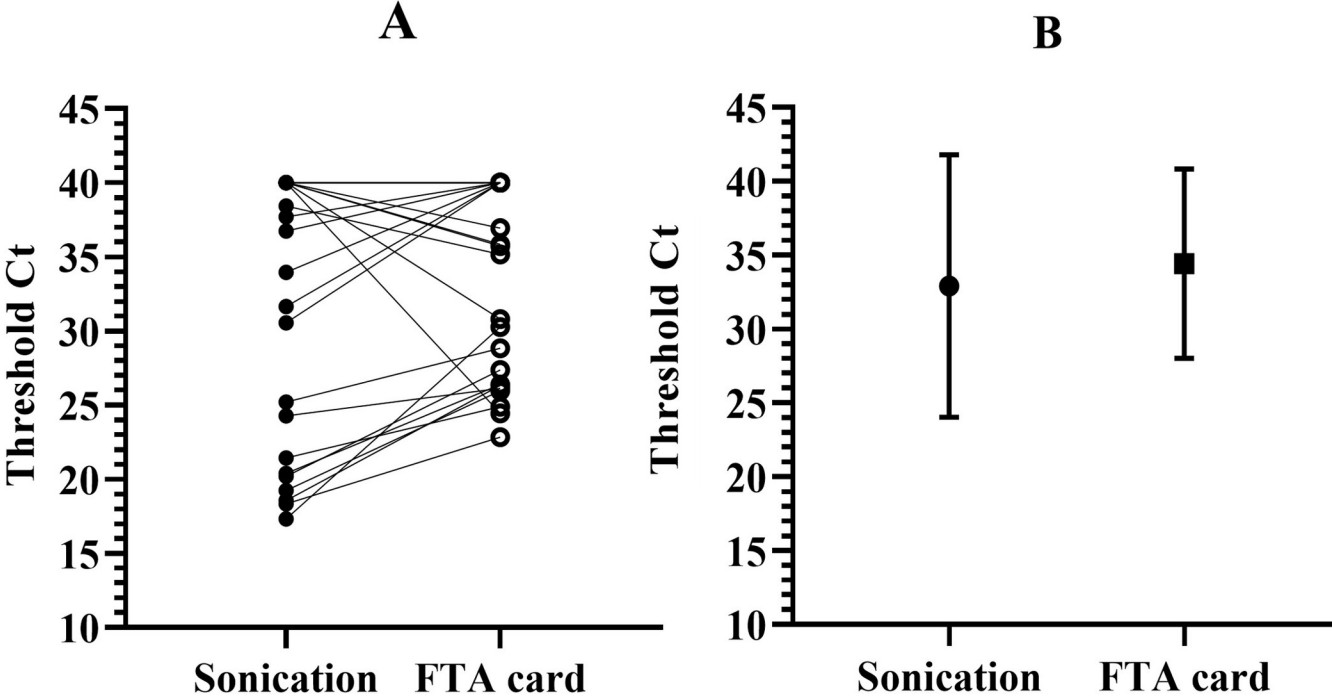

**Fig 2. Comparison of threshold cycles between the sonication and FTA card qPCR protocols for detection of *IS6110*.** Panel A shows the same samples processed by each experimental protocol. qPCR was performed in the standard StepOne instrument. Data is shown as the average threshold Ct for each sample, which was tested in duplicate. Panel B shows mean ± SD for each experimental protocol, averaging all samples.

**Table 1. Cts corresponding to the detection of *IS6110* in DNA extracted by each protocol (sonication or FTA card) from TB-confirmed clinical samples.**

| Sample | Sonication (Ct) | FTA card (Ct) | GeneXpert MTB/RIF | Culture |
|---|---|---|---|---|
| 1 | - | 35.83 | - | - |
| 2 | - | - | - | - |
| 3 | 37.70 | - | - | - |
| 4 | 21.45 | 24.90 | + | + |
| 5 | - | - | - | - |
| 6 | - | - | - | - |
| 7 | - | - | - | - |
| 8 | - | 35.73 | - | - |
| 9 | - | - | - | - |
| 10 | - | 36.95 | - | - |
| 11 | 36.75 | - | - | - |
| 12 | - | - | - | - |
| 13 | - | - | - | - |
| 14 | - | - | - | - |
| 15 | 31.66 | - | - | - |
| 16 | 38.42 | 35.16 | + | - |
| 17 | 19.24 | 25.98 | + | + |
| 18 | 18.57 | 26.32 | + | + |
| 19 | 25.21 | 28.84 | + | + |
| 20 | - | 30.80 | + | + |
| 21 | 33.97 | - | + | - |
| 22 | 30.54 | - | + | - |
| 23 | 20.19 | 27.37 | + | + |
| 24 | - | - | - | - |
| 25 | 18.34 | 22.85 | + | + |
| 26 | 24.28 | 26.14 | + | + |
| 27 | 17.34 | 30.26 | + | + |
| 28 | - | 24.45 | + | + |
| 29 | 20.39 | 26.40 | + | + |

[a]corresponding symbols in the table (+) positive; (-) negative

presented a false positive result by the sonication protocol, while the FTA card protocol showed 100% agreement with the culture (Table 2).

Cohen´s agreement between the results of the experimental protocols (sonication versus FTA card) showed a kappa index of 0.7 (95% CI: 0.69–0.79; "substantial"). The same level of agreement was found when comparing the sonication protocol versus culture (kappa = 0.7; 95% CI: 0.69–0.79). Interestingly, when the FTA card protocol was compared to culture, it showed a kappa index of 1.0 (95% CI: 0.80–0.99), higher than GeneXpert's index of 0.7, thus suggesting a better performance (Table 2). The relationship between the results obtained by each protocol is shown in the Venn diagram in Fig 3 (raw data shown in S1 Table). One sample was detected solely by sonication extraction and GeneXpert MTB/RIF, two samples were detected only by GeneXpert MTB/RIF, and two samples were detected simultaneously by GeneXpert MTB/RIF, culture, and FTA card extraction protocol, while nine samples were detected by all protocols. Sensitivity (SE) and specificity (SP) of the methods were determined in comparison with culture (Table 2). The protocol using extraction by sonication presented a SE of 80% (95% CI: 63–96) and SP of 89% (95% CI: 76–102), while the protocol using FTA card

**Table 2. Sensitivity, specificity, and kappa index of results from both experimental extraction protocols compared to GeneXpert MTB/RIF and culture.**

| | | GeneXpert MTB/RIF | | Culture | | Comparison to culture | | |
|---|---|---|---|---|---|---|---|---|
| | | Positive | Negative | Positive | Negative | Sensitivity | Specificity | Kappa |
| Sonication | Positive | 10 | 0 | 9 | 1 | 80% | 89% | 0,7 |
| | Negative | 4 | 15 | 2 | 17 | (95% CI: 63–96) | (95% CI: 76–102) | (95% CI: 0,6–0,79) |
| FTA card | Positive | 11 | 0 | 11 | 0 | 100% | 100% | 1,0 |
| | Negative | 3 | 15 | 0 | 18 | (95% CI: 98–101) | (95% IC: 98–101) | (95% CI: 0,98–1,01) |
| GeneXpert MTB/RIF | Positive | | | 11 | 3 | 100% | 79% | 0,7 |
| | Negative | | | 0 | 15 | (95% CI: 98–101) | (95% CI: 60–95) | (95% CI: 0,6–0,79) |

presented 100% SE and SP (95% CI: 98–101). The gold standard molecular test GeneXpert MTB/RIF displayed an SE of 100% (95% CI: 98–101) and SP of 79% (95% CI: 60–95).

Since the FTA card protocol detected more positive samples than the sonication protocol and yielded better sensitivity, specificity, and kappa index, it was chosen for the remainder of the present study as a simplified and portable DNA extraction method.

### Evaluation of qPCR with intact *Mtb* cells and colony-forming unit counting

Starting from a *Mtb* suspension corresponding to turbidity 1 on the McFarland scale, colony growth could be observed up to the fifth 1:10 dilution. Counting could be performed up to the fourth and fifth dilutions, yielding an average of 9.5 ± 6.3 and 0.5 ± 0.7 CFU/ml, respectively (Table 3, raw data presented in S2 Table). DNA extracted from each dilution by the simplified FTA card-based protocol was used for qPCR detection of *M. tuberculosis* DNA in the portable Q3-Plus system (Fig 4). The qPCR efficiency was determined to be 118% (slope -2.87 and R2 = 0.95) for Step One, and 216% (slope -1.96 and $R^2$ = 0.99) for the Q3-Plus (Table 3 and

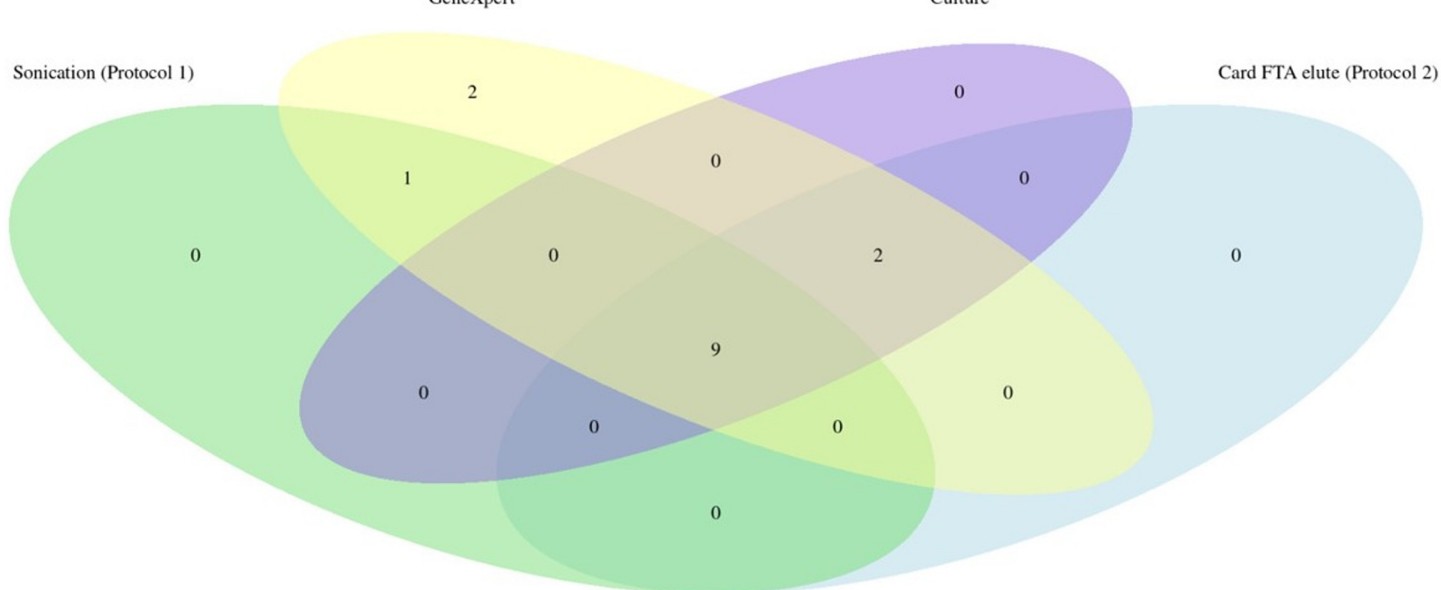

**Fig 3. Venn diagram showing the number of samples detected by each technique.** Numbers in overlapping areas indicate that the techniques agreed with the classification, whether the sample was negative or positive. All four techniques equally detected the presence of MTB DNA in 9 samples, while 2 were detected by FTA card protocol, culture, and GeneXpert MTB/RIF assay. One sample was detected only by the GeneXpert MTB/RIF assay and sonication, and 2 were solely detected by the GeneXpert MTB/RIF assay. Raw data is shown in S1 Table.

**Table 3. Colony forming unit and IS6110 detection by two qPCR instruments using 1:10 dilutions of a *M. tuberculosis* cell suspension.** UNC means "uncountable".

| Diluiton | Average (CFU/mL) | Step One (± SD) | Q3 Plus (± SD) |
|---|---|---|---|
| 1:10 | UNC | 23.7 ± 0.9 | 25.7 ± 2.1 |
| 1:100 | 175 ± 35.3 | 29.4 ± 0.7 | 27.9 ± 0.4 |
| 1:1.000 | 35.5 ± 6.3 | 31.5 ± 0.1 | 29.2 ± 0.8 |
| 1:10.000 | 9.5 ± 6.3 | 33.3 ± 1.1 | 32.1 ± 0.7 |
| 1:100.000 | 0.5 ± 0.7 | 36.1 ± 1.1 | 36.5 ± 1.4 |

S1 Fig). The FTA card protocol presented an apparent $LOD_{95\%}$ of 19.3 CFU/mL of MTB when the portable Q3-Plus instrument was used (Fig 4 and S3 Table). We were not able to calculate the $LOD_{95\%}$ when the StepOne instrument was used because all replicates were positive.

Next, we validated the completed portable solution (DNA extraction + qPCR) using 98 samples in parallel to the routine diagnostic service. Sputum samples were processed by the FTA card protocol and evaluated for the presence of *Mtb* DNA in the portable and in benchtop instruments (Fig 5) shows representative curves of IS6110 and 18S rRNA detection as detected by Q3-Plus using DNA extracted from sample #34. Of the 98 samples, 34 (34,6%) were positive for *Mtb* DNA in the Q3-Plus and 45 (45,9%) were positive on GeneXpert Ultra, with 27 (27,5%) samples being detected by both instruments.

We then produced a ROC curve comparing the results obtained by the portable platform (FTA card protocol followed by qPCR in the Q3-Plus) and the GeneXpert MTB/RIF system against the classification of samples by TB case, the WHO reference standard method. The plots yielded an area under the curve (AUC) of 0.78 for the portable testing solution and 0.93 for the GeneXpert Ultra assay. As p<0.001, it implies that these results are statistically significant (Fig 6).

The Q3 Plus provides on its platform amplification results with "threshold" (detectable fluorescence level) and "baseline" (defined limit of PCR cycles) parameters used to determine the Ct (cycle threshold) of each sample as shown in Fig 4. Based on our results, we propose a classification algorithm that defines Ct <34 as positive, 34<Ct>35 indicates a retest, and Ct>36 as negative. On the other hand, the GeneXpert Ultra results are provided as five semi-quantitative detection levels: trace, very low, low, medium, and high. S4 Table shows the

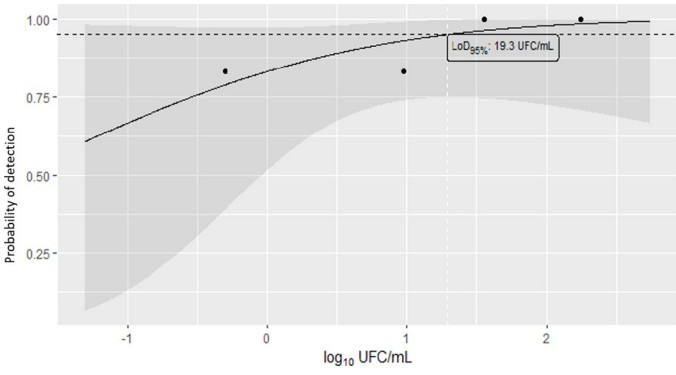
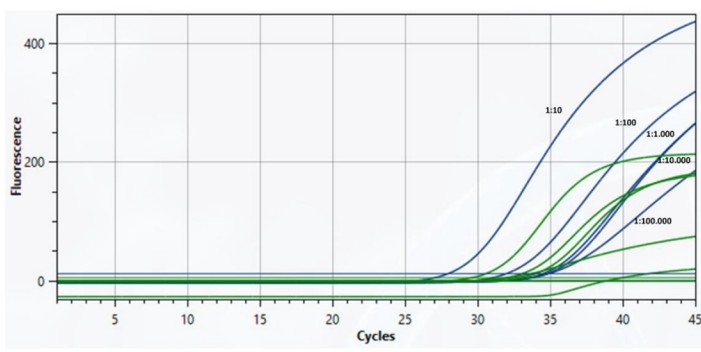

**Fig 4. Representative traces for the detection of *Mtb* DNA by the portable qPCR instrument Q3-Plus and its calculated limit of detection probability as measured by colony forming units (CFU).** Panel A. Shown are traces for the detection of the *Mtb* IS6110 genomic marker in the Q3-Plus qPCR system, using DNA extracted by the FTA card experimental protocol from dilutions 1:10, 1:100, 1:1.000, and 1:10.000 dilutions. Such traces were used to calculate the data shown in Table 3 (see also S2 Table). Each sample was run in duplicate. Panel B. A Probit regression analysis was performed on the probability of detection of each *M. tuberculosis* concentration by the Q3-Plus system, as shown in Table 3 (S2 and S3 Tables), yielding a $LOD_{95\%}$ of 19.5 CFU/mL.

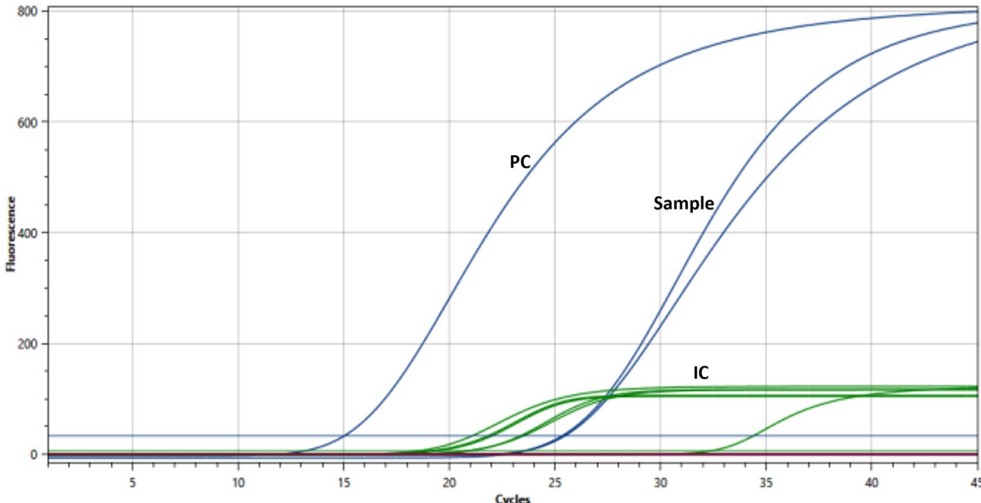

**Fig 5. Representative traces of qPCR detection of *M. tuberculosis* IS6110 and human 18S rRNA obtained from a human sample.** Representative traces of qPCR detection of the *Mtb* genomic marker *IS6110* (blue lines) and the human 18S rRNA (green lines, "IC") for sample #34 ("sample") as well as the positive control H37Rv DNA (10 ng/μL, "PC"). Each sample was tested in duplicate.

correlation between the results of the Q3 Plus (positive or negative, based on the above algo-rithm) and the semiquantitative results provided by GeneXpert Ultra. The Q3 plus instrument detected 34 positive samples, which presented the following results on the GeneXpert Ultra: 7 negatives, 1 positive trace, 1 positive very low, 1 positive low, 4 positives medium, and 20 posi-tives high. A total of 36 samples were negative, showing the following results on the GeneX-pert: 30 negatives, 3 positive trace, 1 positive very low, 2 positives low, 0 positive medium, and

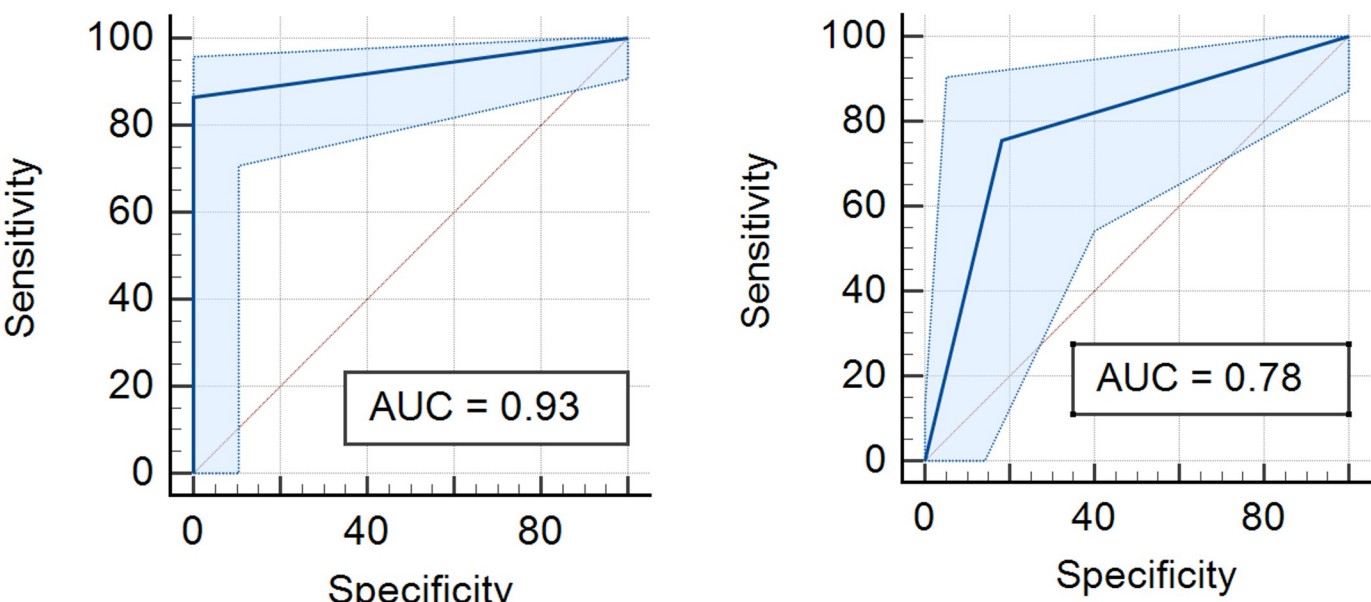

**Fig 6. ROC curve for the detection of *IS6110 Mtb* marker using the Q3-Plus and FTA card protocol or the GeneXpert Ultra system versus culture.** Results obtained after the analyses of all samples by both molecular methods are plotted against the sample classification as shown by culture, the gold standard method. Diagonal segments were produced by bonds.

0 positive high. Finally, the 28 samples that were considered as retests on the Q3 plus presented the following results on the GeneXpert: 16 negatives, 2 positives traces, 3 positives very low, 2 positives low, 2 positives medium, and 3 positives high, as shown in (S4 Table). When analyzing the agreement of molecular methods with culture, we obtained a substantial agreement (k = 0.62) for Q3 Plus and a fair agreement (k = 0.31) for GeneXpert Ultra, with sensitivity of 92% and 88% and specificity of 61% and 62%, respectively for each instrument. When analyzing the agreement of the molecular methods with the TB case classification, the portable protocol (FTA card and Q3-Plus) presented a moderate agreement (k = 0.57) and GeneXpert Ultra showed a fair agreement (k = 0.32), with a sensitivity and specificity respectively of 75% and 81% and GeneXpert Ultra 86% and 100%. The results obtained by both instruments, GeneXpert Ultra and Q3-Plus, showed a substantial agreement (k = 0.62). The positivity by all four methods and their convergence is shown in Fig 7 (raw data shown in S5 Table). Of the 98 samples, 12 samples were positive by all four diagnostic techniques (culture, case TB, GeneXpert Ultra, FTA card, and Q3-Plus instrument), 15 were positive by the Q3-Plus system, GeneXpert and case TB, 01 samples was positive by case TB and culture, 06 were positive by GeneXpert and case TB, 02 were positive only by case TB, and 01 sample was positive by the portable platform and case TB. Finally, all 12 samples identified as positive by culture were also positive by the portable method, and of the 34 samples identified as positive by the Q3-Plus instrument, 28 were also positive by case TB, while all 45 samples positive by the GeneXpert Ultra assay were positive by case TB.

## Discussion

Tuberculosis remains a significant problem, particularly in countries with high poverty rates and areas with limited access to healthcare and infrastructure [23]. In this study, we evaluated a molecular assay to aid in the screening and detection of TB using a portable system with potential point-of-care capabilities. The use of portable platforms represents a significant advance in diagnostic technology [24]. Our study used a simplified DNA extraction protocol

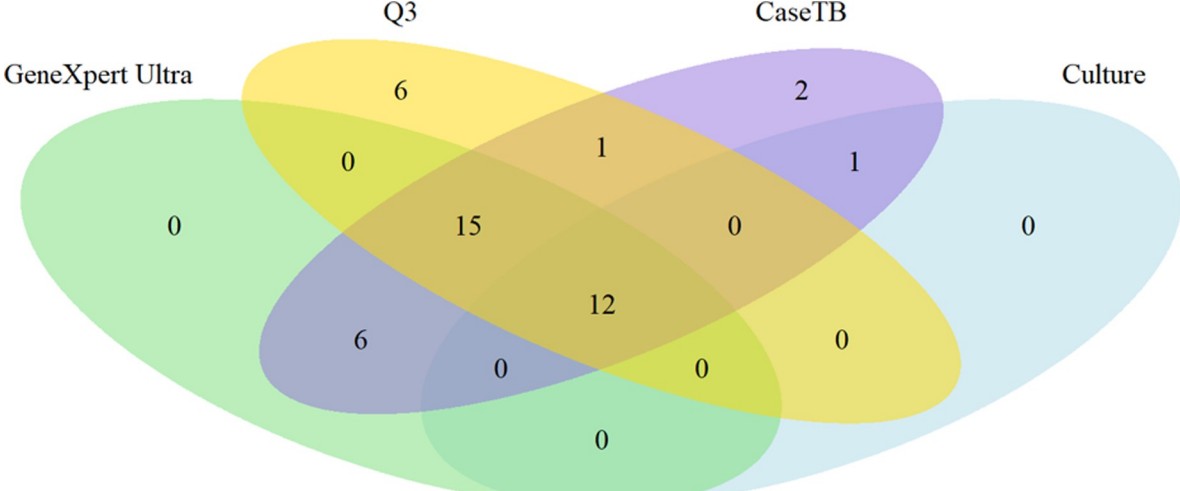

**Fig 7. Venn diagram showing the relationship between and the number of samples that were detected by each technique.** TB clinical case classification indicated 52 samples as positive, while 45 samples were detected as positive only in GeneXpert and TB Case, 28 samples were positive only in Q3 and TB case classification, 15 samples were detected simultaneously by GeneXpert Ultra and culture, 17 were detected by the portable protocol and culture, and 12 samples were detected by all protocols. Out of the diagram, 46 samples were deemed negative and 28 were undetermined. Raw data shown in S5 Table.

previously shown to liquefy the sample and to facilitate the retention of the DNA in the FTA card, which is essential for sample storage and transport [14]. FTA cards have already been used in studies with sputum, blood, and saliva samples [13, 25–27]. Efficient nucleic acid extraction is essential for the successful diagnosis of TB by qPCR and for effectively contributing to disease control [28]. Another tool for disease control is to increase the population's access to molecular tests, which is hampered by the difficulties of using and maintaining the technique's sensitive instruments, such as centrifuges and thermocyclers, in areas with limited or non-existent laboratory infrastructure [29].

In the Brazilian Unified Health System (SUS), the GeneXpert test is implemented for routine TB diagnosis and rifampin resistance detection [8]. Since this instrument requires a robust laboratory infrastructure, its use is impaired by the logistical difficulties of hard-to-reach areas. To overcome these obstacles, we sought to validate a simplified DNA extraction method from sputum that could be used in low infrastructure setting or point of care situations, since a portable qPCR instrument has already been developed [16].

In this study, two simple methods for DNA extraction from sputum were evaluated. The first is based on sonication, which is a common in-house method where cell membrane rupture occurs through ultrasound. The second method uses a strong denaturing chemical and proteinase K followed by solubilization by detergents embedded in the FTA Elute Micro Card to release the bacilli DNA [8, 14]. The two simplified DNA extraction protocols showed an important agreement (70%), with eleven out of 29 samples testing positive by qPCR with DNA obtained by FTA card protocol, and nine out of 29 samples testing positive with DNA obtained by sonication. Three out of the 29 samples showed divergence in detection by qPCR relative to the GeneXpert assay classification. Two true positives samples were detected by FTA card extraction and positive by culture and GeneXpert MTB/RIF were negative by the sonication extraction method. These divergences may occur due to the breaking of genomic DNA in the sonication process. Mandakhalikar et al. (2018) also used the sonication DNA extraction method and reported that high temperatures in the protocol can hinder DNA recovery [30]. In addition, the cavitation in the water bath sonicator is low, making the process uneven [30]. Accordingly, a positive sample by the FTA card protocol and positive by GeneXpert MTB/RIF but negative by sonication may have been a function of sample heterogeneity or the low presence of the genomic targets (low bacterial index). Interestingly however, discordant samples #20, #22, and #28 show high Ct in the qPCR, suggesting that some of the above considerations might play a role in the discrepant results [30, 31]. Using a smaller volume for elution in both protocols could potentially yield better results, but this was not evaluated. The comparative evaluation of the results of the two extraction protocols followed by qPCR in a standard instrument showed a strong agreement with the results obtained by the GeneXpert MTB/RIF assay, with disagreement between only two samples (#16 and #21, both negative by standard qPCR and positive by GeneXpert MTB/RIF). The two samples were also culture negative, suggesting that they are false positives. This is suggested by the low specificity of 79% when GeneXpert MTB/RIF was compared to the culture. Even considering a sensitivity of 100%, such lack of specificity can influence the diagnosis of negative cases, impacting clinical outcomes. It is worth noting that GeneXpert MTB/RIF, like any qPCR technique, equally amplifies DNA fragments from alive and dead bacilli (that is, viable or not for growth in culture), thus sometimes producing "false positive results" compared to culture. Indeed, Silva et al. (2019) [32] showed that patients undergoing treatment or with a history of positive TB can be qPCR positive for up to five years, highlighting the possible low specificity of such tests [32]. When the results of the sonication protocol were compared with GeneXpert MTB/RIF, there was inconsistency in four samples that were false negatives. Of these four, two were also culture positive. The

relationship between the detection performance of each technique can be visualized in a Venn diagram (Fig 7).

Results obtained by the FTA card protocol and the portable qPCR instrument Q3-Plus showed perfect sensitivity (100%) and specificity (100%) when compared to culture, in agreement with previous data from our group. Ali and colleagues (2020) also used FTA Elute Micro cards, and the results showed almost perfect agreement in sensitivity and specificity when compared with GeneXpert MTB/RIF and culture, with only one false positive [14]. qPCR results obtained by the sonication protocol showed less sensitivity and specificity in relation to culture (80% and 89%). Interestingly, specificity was slightly greater than that of GeneXpert MTB/RIF. However, issues related to false positive and false negative results would require improvements in the sonication protocol to produce more accurate and acceptable results.

According to the WHO [33], greater sensitivity and low specificity are recommended in screening tests, as in the case of the portable POC platform described in the present study. Although the sonication protocol uses commonly available reagents and common laboratory equipment, it requires washing steps just like commercial kits, resulting in a longer and more laborious procedure. As for the limit of detection (LOD), it provided a good detection capacity (up to 19.3 CFU/mL). The $LOD_{95\%}$ obtained in our study was in the same range as that of Chakravorty et al. (2017), who analyzed the performance of the GeneXpert MTB/ Ultra as a point-of-care (POC) assay suitable for diagnosing TB and obtained a $LOD_{95\%}$ of 15.6 CFU/mL [34]. The FTA card protocol, however, showed greater sensitivity and specificity, as well as other benefits such as easiness of operation (a lesser requirement for laboratory equipment, for example), transportation, storage, and conservation of the genetic material [35]. Evaluation of the DNA amplification with the Q3-Plus system showed fair agreement (Kappa = 0.62) with the GeneXpert Ultra assay, which was more sensitive than GeneXpert when compared to culture. Despite both methods being molecular, the Q3-Plus detected more cases than GeneXpert Ultra. Despite the high sensitivity in our study, the specificity was low, which could be justified by the medical history of the patients, where some were already in treatment, consistent with the studies by Zhang et al. 2021 and Silva et al. 2019 [32, 36]. Zhang and colleagues, as well as Silva and colleagues, highlighted in their studies that the decrease in specificity is related to the previous history of TB, and molecular methods can detect patients in treatment, i.e., with dead bacilli, justifying the low specificity (61% in our study with 22 false positives). It is known that only live bacilli grow in culture, and these studies also emphasize the importance of knowing the patient's history, especially if they are in treatment, along with the molecular diagnosis [32, 36]. Our results provide evidence, however, that the simplified DNA extraction protocol followed by qPCR detection in the Q3-Plus system can detect *M. tuberculosis* DNA at all stages of the disease, thus being a suitable tool for TB screening at the point of care. Interestingly, a modelling study on the estimated impact of implementing the portable GeneXpert Ultra assay concluded that more sensitive molecular assays should be used in active searches during site-specific implementation, prioritizing populations with anticipated high prevalence of TB [37]. We propose the same use for the simplified portable protocol shown in the present study.

In our study, the GeneXpert assay showed fair agreement with 30 false positives and a specificity of 62% when compared to culture, the gold standard and the Q3 plus assay showed fair agreement with 22 falses positives and specificity 61% when compared to culture, the gold standard. The impact of false negative and false positive results in the health system have been extensively studied for tuberculosis and its known antibiotic resistance. The main impact of false negative tests is the continuation of community transmission by the untreated patient. On the other hand, the main impact of false positive results is the unnecessary treatment of the patient, increasing the chances of antibiotic resistance due to unnecessary use of such drugs. However, one might not forget that the physician's clinical evaluation should always prevail

over the results of a molecular test [38–41]. This is especially true in the case of screening tests, such as the one validated in the present study.

Although the GeneXpert Ultra assay showed better results and a faster turnaround time of 80–90 minutes (with about 20 minutes of hands-on time), it has several drawbacks: it is expensive without subsidies (in Brazil, US$30 after subsidies, and US$180 without subsidies) [42], each cartridge processes only one sample, and the system is not suitable for areas with limited infrastructure. In contrast, the portable platform would cost around US$40 without subsidies, it does not consume all sample's volume thus allowing retesting, it might be used in limited infrastructure situations, it processes two samples per run and provides results in up to 4 hours (about 20–25 minutes of hands-on time). Additionally, preliminary data show that the portable platform might be transported to remote health units or regions where the only available infrastructure is electricity and a roof, thus aiding in the screening of suspected TB cases and effectively helping to contain the spread of the disease in local communities.

At its current stage of prototype validation, the simplified protocol and the user interface for qPCR result interpretation are not optimized for use by unskilled technicians. We are currently working together with the health agents who would be using the platform to overcome these aspects. However, it is important to highlight that the portable platform is designed for screening of suspected cases of TB during localized active searches, not for routine diagnosis. Therefore, we understand that the portable system should be used by minimally trained technicians, which should be able to perform the few steps of the simplified protocol and qPCR run, considering that data analysis is easily automated.

## Conclusion

The present study validated a simplified and portable DNA extraction procedure from sputum samples, suitable for point-of-care qPCR screening of individuals suspected of TB infection. We suggest that the simplified extraction protocol coupled with a portable qPCR instrument might be used in low-infrastructure settings or in areas where people lack access to healthcare centers or are highly vulnerable (such as prisons), thus effectively making molecular TB screening accessible to everyone.

## Supporting information

**S1 Table. Results table for Venn diagram construction calculations.**
(PDF)

**S2 Table. Data used to calculate the averages presented on Table 3.**
(PDF)

**S3 Table. Detection probability Q3 plus and step one.**
(PDF)

**S4 Table. Correlation between Q3 plus and GeneXpert Ultra semiquantitative results.**
(PDF)

**S5 Table. Results table for Venn diagram construction calculations.**
(PDF)

**S1 Fig. Plotting graphs to determine the efficiency of Q3 plus and step one.**
(PDF)

## Acknowledgments

We are greatful to the Department of Tisiology and Leprosy of the Health Department at the University Hospital of Canoas (Canoas, Brazil), for their support in obtaining the samples.

## Author Contributions

**Conceptualization:** Maria Lucia Rossetti, Alexandre Dias Tavares Costa.

**Data curation:** Tainá dos Santos Soares, Regina Bones Barcellos, Maria Lucia Rossetti.

**Formal analysis:** Tainá dos Santos Soares, Alexandre Dias Tavares Costa.

**Funding acquisition:** Maria Lucia Rossetti, Alexandre Dias Tavares Costa.

**Investigation:** Tainá dos Santos Soares, Graziele Lima Bello, Ianca Moraes dos Santos Petry, Maria Rita Castilho Nicola, Larissa Vitoria da Silva, Joana Morez Silvestri.

**Methodology:** Tainá dos Santos Soares, Graziele Lima Bello, Ianca Moraes dos Santos Petry, Maria Rita Castilho Nicola, Larissa Vitoria da Silva, Regina Bones Barcellos, Alexandre Dias Tavares Costa.

**Project administration:** Maria Lucia Rossetti, Alexandre Dias Tavares Costa.

**Resources:** Regina Bones Barcellos, Joana Morez Silvestri, Maria Lucia Rossetti, Alexandre Dias Tavares Costa.

**Supervision:** Regina Bones Barcellos, Maria Lucia Rossetti, Alexandre Dias Tavares Costa.

**Validation:** Regina Bones Barcellos, Joana Morez Silvestri, Alexandre Dias Tavares Costa.

**Writing – original draft:** Tainá dos Santos Soares, Alexandre Dias Tavares Costa.

**Writing – review & editing:** Tainá dos Santos Soares, Maria Lucia Rossetti, Alexandre Dias Tavares Costa.

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
