## [Decision Letter · Decision Letter 0]

19 Jun 2024

PONE-D-24-12276Laboratory validation of a simplified DNA extraction protocol followed by a portable qPCR detection of M. tuberculosis DNA suitable for point of care settings.PLOS ONE

Dear Dr. Costa,

Thank you for submitting your manuscript to PLOS ONE. After careful consideration, we feel that it has merit but does not fully meet PLOS ONE’s publication criteria as it currently stands. Therefore, we invite you to submit a revised version of the manuscript that addresses the points raised during the review process.

Thank you for submitting your well written manuscript to this journal. We believe that the comments from the reviewers are very constructive, should be fairly easy to respond to, and will improve the overall impact of the manuscript. We look forward to receiving the revised manuscript soon.

We look forward to receiving your revised manuscript.

Kind regards,

Shahriar Ahmed, MBBS, MHE, MPhil

Academic Editor

PLOS ONE

Additional Editor Comments:

Thank you for submitting your well written manuscript to this journal. We believe that the comments from the reviewers are very constructive, should be fairly easy to respond to, and will improve the overall impact of the manuscript. We look forward to receiving the revised manuscript soon.

Reviewers' comments:

Reviewer's Responses to Questions

**Comments to the Author**

1. Is the manuscript technically sound, and do the data support the conclusions?

Reviewer #1: Yes

Reviewer #2: Yes

2. Has the statistical analysis been performed appropriately and rigorously? 

Reviewer #1: Yes

Reviewer #2: Yes

3. Have the authors made all data underlying the findings in their manuscript fully available?

Reviewer #1: Yes

Reviewer #2: Yes

4. Is the manuscript presented in an intelligible fashion and written in standard English?

Reviewer #1: Yes

Reviewer #2: Yes

5. Review Comments to the Author

Reviewer #1: In the introduction

Please update data in the introduction (2021, 2022) with last up to date data if available.

Smear microscopy has low sensitivity but quite good specificity.

Real time PCR section

Explain why IS6110 region was selected to detect MTB.

You are using the Xpert Ultra cartridge. It is important to specify that from the beginning of the paper, and outlining its main difference with the regular MTB/Rif cartridge in terms of specificity and sensitivity

In the discussion.

It would be interesting to give a rough idea of how it would cost to test one individual when using the FTA + Q3-Plus. Obviously not doing a deep cost-effectiveness analysis but show how many individuals can be tested under routine practice by Xpert compared to FTA + Q3-plus.

I would recommend address more in details the trace results and elaborate more on the comparison/correlation of trace results vs FTA + Q3 plus.

In remote areas, it is usually the case that the level of competency of laboratory technicians is more limited. I suggest to mention how feasible it would be to implement this technology especially around the results interpretation, is there any language barrier to read the results,.

Please elaborate on the easiness for operating and what it really means for the laboratory technician if s/he were to use this technology. You mention it takes up to 4 hours using Q3-plus. I guess you are also including the extraction steps in those 4 hours, please specify in your text. Also please add details on how long the lab technician needs to have hands-on for samples handling and processing. For example for GeneXpert it could around 20 minutes hands-on (adding buffer, mixing for 10 + 5 minutes, load the sample in the cartridge, launch the test). Please indicate out of the 4 hours, how long is required for actual hands-on.

Please add some details on what it implies for the health system to end up with false negative and/or false positive results.

Please elaborate on the testing capacity of Xpert vs FTA+Q3-plus. How many samples can be tested in one run of 4 hours?

Reviewer #2: Current manuscript has described laboratory validation of a simplified DNA extraction protocol as a POC. Few issues that need to be discussed in the manuscript.

1. How affordable the assay will be as POC?

2. Have you compared the sensitivity, specificity of the children's specimens? Do you think this can be different?

3. Why only adults specimens had been chosen?

Thank you.

6. PLOS authors have the option to publish the peer review history of their article (what does this mean?). If published, this will include your full peer review and any attached files.

Reviewer #1: No

Reviewer #2: No

---

## [Author Response · Author response to Decision Letter 0]

6 Sep 2024

Reviewer #1

In the introduction

1. Please update data in the introduction (2021, 2022) with last up to date data if available.

Reply: Data in the introduction section was updated. (lines 29,30 and 31)

2. Smear microscopy has low sensitivity but quite good specificity.

Reply: The information is clarified (lines 33,34,35,36 and 37). 

3. Explain why IS6110 region was selected to detect MTB.

Reply: The IS6110 was selected because it is a specific marker of the MTB complex (20, 21)and it used in several commercial diagnostic tests for tuberculosis. The explanation is stated in lines 151, 152, 153 and 154. 

4. You are using the Xpert Ultra cartridge. It is important to specify that from the beginning of the paper, and outlining its main difference with the regular MTB/Rif cartridge in terms of specificity and sensitivity

Reply: We have specified all instances where Xpert Ultra was used. We also outlined the differences between the regular MTB/RIF and Ultra cartridges (lines 46-54). 

In the discussion.

5. It would be interesting to give a rough idea of how it would cost to test one individual when using the FTA + Q3-Plus. Obviously not doing a deep cost-effectiveness analysis but show how many individuals can be tested under routine practice by Xpert compared to FTA + Q3-plus.

Reply: Although the GeneXpert Ultra assay showed better results and a faster turnaround time of 80–90 minutes (with about 20 minutes of hands-on time), it has several drawback: it is expensive without subsidies (in Brazil, US$30 after subsidies, and US$180 without subsidies) [42], each cartridge processes only one sample, and the system is not suitable for areas with limited infrastructure. In contrast, the portable platform would cost around US$40 without subsidies, it does not consume all sample’s volume thus allowing retesting, it might be used in limited infrastructure situations, it processes two samples per run and provides results in up to 4 hours (about 20-25 minutes of hands-on time). Additionally, preliminary data show that the portable platform might be transported to remote health units or regions where the only available infrastructure is electricity and a roof, thus aiding in the screening of suspected TB cases and effectively helping to contain the spread of the disease in local communities.

6. I would recommend address more in details the trace results and elaborate more on the comparison/correlation of trace results vs FTA + Q3 plus.

Reply: 

7. In remote areas, it is usually the case that the level of competency of laboratory technicians is more limited. I suggest to mention how feasible it would be to implement this technology especially around the results interpretation, is there any language barrier to read the results,.

Reply: At the current stage of development (validation of a prototype), we have not yet dealt with language barriers during implementation of the test. However, we do expect that a user friendly interface is created before the product launch in the market. This information is included in lines 498 -505. 

8. Please elaborate on the easiness for operating and what it really means for the laboratory technician if s/he were to use this technology. You mention it takes up to 4 hours using Q3-plus. I guess you are also including the extraction steps in those 4 hours, please specify in your text. Also please add details on how long the lab technician needs to have hands-on for samples handling and processing. For example for GeneXpert it could around 20 minutes hands-on (adding buffer, mixing for 10 + 5 minutes, load the sample in the cartridge, launch the test). Please indicate out of the 4 hours, how long is required for actual hands-on.

Reply: The simplified protocol requires less than 30 minutes of hands-on time per 4-hour run. Our preliminary data shows that the simple steps of our portable protocol are easily performed by minimally trained technicians, which is the intended public for using it. This information is stated in lines 137-149/ 164-177/ 492 and 493. 

9. Please add some details on what it implies for the health system to end up with false negative and/or false positive results.

Reply: The implications of false positive/negative results to the health system is stated in lines 477-485. 

10. Please elaborate on the testing capacity of Xpert vs FTA+Q3-plus. How many samples can be tested in one run of 4 hours?

Reply: Both instruments have similar capacities. Two samples can be tested in one run of 4 hours in the FTA-Q platform, while one sample can be analyzed per Xpert cartridge. However, the Ultra system can run up to 4 cartridges, while the Q3 software can command up to 6 instruments, where each instrument would run two samples per 4-hour turn. This information is stated in lines 486-497. 

Reviewer #2: 

1. How affordable the assay will be as POC?

Reply: The simplified extraction protocol is very cheap (around US$ 1-2 per sample). The portable qPCR reagents, especially the silicon chip, are expensive (around US 30 per sample). This is one of the main reasons why we suggest that the portable platform (DNA extraction + qPCR) should be used in specific activities, geared towards high prevalence populations in remote or low-resource areas (lines 486-488). 

2. Have you compared the sensitivity, specificity of the children's specimens? Do you think this can be different?

Reply: We have not compared the sensitivity and specificity of the simplified protocol with children´s samples. This information is not state in lines 106. However, we do not anticipate any differences between adult and children’s samples besides those already described in the literature, i.e., lower sensitivity due to the difficulty in obtaining a good sample from infant patients. 

3. Why only adults specimens had been chosen?

Reply: The main reason was due to the easiness of collection, since children often are not able to produce a sputum sample.

---

## [Decision Letter · Decision Letter 1]

9 Oct 2024

Laboratory validation of a simplified DNA extraction protocol followed by a portable qPCR detection of M. tuberculosis DNA suitable for point of care settings.

PONE-D-24-12276R1

Dear Dr. Costa,

We’re pleased to inform you that your manuscript has been judged scientifically suitable for publication and will be formally accepted for publication once it meets all outstanding technical requirements.

Kind regards,

Shahriar Ahmed, MBBS, MHE, MPhil

Academic Editor

PLOS ONE

Additional Editor Comments (optional):

Thank you for submitting a revised version addressing all the comments from reviewers comprehensively. I would like to take this opportunity to congratulate all the authors. Thank you for submitting your manuscript to our journal and we look forward to working with you again in future.

Reviewers' comments:

Reviewer's Responses to Questions

**Comments to the Author**

1. If the authors have adequately addressed your comments raised in a previous round of review and you feel that this manuscript is now acceptable for publication, you may indicate that here to bypass the “Comments to the Author” section, enter your conflict of interest statement in the “Confidential to Editor” section, and submit your "Accept" recommendation.

Reviewer #1: All comments have been addressed

Reviewer #2: All comments have been addressed

2. Is the manuscript technically sound, and do the data support the conclusions?

Reviewer #1: Yes

Reviewer #2: Yes

3. Has the statistical analysis been performed appropriately and rigorously? 

Reviewer #1: Yes

Reviewer #2: Yes

4. Have the authors made all data underlying the findings in their manuscript fully available?

Reviewer #1: Yes

Reviewer #2: Yes

5. Is the manuscript presented in an intelligible fashion and written in standard English?

Reviewer #1: Yes

Reviewer #2: Yes

6. Review Comments to the Author

Reviewer #1: Many thanks for all the adjustments. All concerns were addressed.

Reviewer #2: Thank you for addressing and incorporating all the responses in the manuscript. This looks good now for going forward for publication.

7. PLOS authors have the option to publish the peer review history of their article (what does this mean?). If published, this will include your full peer review and any attached files.

Reviewer #1: **Yes: **Dr Vibol IEM

Reviewer #2: No

---

## [Editor Report · Acceptance letter]

5 Nov 2024

PONE-D-24-12276R1 

PLOS ONE

Dear Dr. Costa, 

I'm pleased to inform you that your manuscript has been deemed suitable for publication in PLOS ONE. Congratulations! Your manuscript is now being handed over to our production team.

Kind regards, 

on behalf of

Dr. Shahriar Ahmed 

Academic Editor

PLOS ONE